# A Label-Free Optical Detection of Pathogens in Isopropanol as a First Step towards Real-Time Infection Prevention

**DOI:** 10.3390/bios11010002

**Published:** 2020-12-23

**Authors:** Julie Claudinon, Siegfried Steltenkamp, Manuel Fink, Taras Sych, Benoît Verreman, Winfried Römer, Morgan Madec

**Affiliations:** 1Faculty of Biology, University of Freiburg, 79104 Freiburg, Germany; julie.claudinon@gmail.com (J.C.); manuel.fink8@googlemail.com (M.F.); sych59@gmail.com (T.S.); winfried.roemer@bioss.uni-freiburg.de (W.R.); 2Signalling Research Centres BIOSS and CIBSS, University of Freiburg, 79104 Freiburg, Germany; ssteltenkamp@ophardt.com; 3Ophardt Hygiene-Technik GmbH + Co. KG, 47661 Issum, Germany; benoit.verreman@etu.unistra.fr; 4Telecom Physique Strasbourg, University of Strasbourg, 67000 Strasbourg, France; 5ICube Laboratory (UMR 7357), CNRS, University of Strasbourg, 67000 Strasbourg, France

**Keywords:** pathogens, optical detection, microscopy, image processing, classification, machine learning

## Abstract

The detection of pathogens is a major public health issue. Every year, thousands of people die because of nosocomial infections. It is therefore important to be able to detect possible outbreaks as early as possible, especially in the hospital environment. Various pathogen detection techniques have already been demonstrated. However, most of them require expensive and specific equipment, and/or complex protocols, which, most of the time, involve biochemical reaction and labelling steps. In this paper, a new method that combines microscopic imaging and machine learning is described. The main benefits of this approach are to be low-cost, label-free and easy to integrate in any suitable medical device, such as hand hygiene dispensers. The suitability of this pathogen detection method is validated using four bacteria, both in PBS (Phosphate Buffered Saline) and in isopropanol. In particular, we demonstrated an efficient pathogenic detection that is sensible to changes in the composition of a mixture of pathogens, even in alcohol-based solutions.

## 1. Introduction

Nosocomial infections, i.e., infections related to hospitalisation, represent a big challenge for the health care system. Hygiene experts estimate that in Germany 400,000 to 600,000 patients per year suffer from a nosocomial infection, with 10,000 to 15,000 of these cases being fatal [1]. In general, up to 30% of all nosocomial infections are considered avoidable [2]. The greatest avoidance potential is found in adequate hand hygiene [3,4,5]. A general, comprehensive screening of patients, visitors and staff for pathogens, allowing the early detection of particular germs, could form an important basis for keeping infections at bay. Therefore, we set up an ambitious project aiming at developing an integrated pathogen detection system coupled with a hand disinfection device, collecting the residual solution from hand-washing and analysing it. Such a system, once developed, will be a continuous monitoring system which will be integrated directly into the hand hygiene procedures, i.e., it will not require any additional gestures or time for the user. It will not aim at replacing standard and quantitative tests but will inform the user in real-time about potential infections, asking them to seek for further assistance if necessary. The work described in this paper focuses on a label-free optical pathogen detection, which is a first breakthrough in this project.

Current pathogen detection methods are relatively slow. Common approaches include microbial cultivation, immunological assays (antibody-based) and molecular detection with amplification methods such as PCR [6,7]. The gold standard methods are still conventional culture-based assays, which are time-consuming and labour-intensive. Cultivation is followed by phenotypic identification, which relies on a combination of biochemical properties such as oxygen requirements, Gram staining, carbohydrate metabolism and the presence of specific enzymes [6]. However, phenotypic identification systems are costly and, again, time-consuming.

Immunological assays (immunochromatographic assays, ELISAs) find their utility in only a limited number of infectious diseases and require specific labels [8,9]. Among molecular detection methods, PCR is the most widely and routinely used for a wide variety of bacteria and viruses. It is fast and precise but needs specialised personnel and equipment. In the past years, RT-PCR became the method of choice [6].

The past decade has also seen the emergence of MALDI-TOF (Matrix Assisted Laser Desorption Ionization – Time of Flight) mass spectrometry, which can identify bacterial isolates within a few minutes and for low costs. This method relies on the comparison of the mass spectrum of the isolate with databases. The method also operates with positive blood cultures and urine samples. However, it requires an upstream isolation and cultivation step and is not very efficient for the identification of species among mixed populations [10]. Additionally, the required equipment is expensive and large, making it difficult to use for smaller facilities or in situ in hospitals.

In addition, miniaturised biosensors have been developed during the past years. They offer several advantages over conventional methods, including high-throughput screening, low limit of detection, real-time analysis and small sample volume. Such pathogen detecting biosensors rely on different technologies including electrochemical (reviewed in [11]), mechanical (reviewed in [12]), nuclear magnetic resonance [13] and optical-sensing methods (reviewed in [14]).

Electrochemical sensors often use voltammetric, amperometric or impedimetric methods with functionalised electrodes with a variety of biochemical compounds ranging from aptamers to antibodies. Upon binding of the target to the electrode, these sensors detect a change in voltage, current or impedance depending on the kind of system used. The electrodes can be very specific to the target analyte. Thus, no labelling is required, but in counterparts, a given electrode can only be used for the intended pathogen [15]. On the other hand, optical pathogen identification systems often rely on colorimetric, fluorescent or chemiluminescent methods. Most function by affinity capture of the analyte and subsequent binding of a labelling reagent for visualisation.

In recent years, plasmonic biosensors are more and more commonplace. These sensors use electromagnetic oscillations on the surface of metals that are excited by light to detect interactions with the analyte. Sensors using localised surface plasmon resonance (LSPR), i.e., the oscillations are located at a single metal nanoparticle [16], or surface-enhanced Raman scattering (SERS), which uses the inelastic scattering of electrons to determine a characteristic spectrum [17], are applied for pathogen identification.

Most of the existing sensors require labelling, a process that is time-consuming, expensive and restricts the number of identifiable pathogens. Commonly used methods in label-free biosensors include plasmonic and electrochemical methods. These methods do not need labels to produce a signal and measure interaction with a functionalised surface (SPR), specific Raman spectra (SERS), or intrinsic electrochemical properties (electrochemical methods) instead. However, even in the absence of labelling, all common biosensors for pathogen recognition still require an affinity capture step [18]. Although it helps increase sensitivity, this is still limiting the identification spectrum as each pathogen requires its own sensor [10].

Another important aspect of bacterial detection systems is their multiplexing capability, i.e., the capacity to detect multiple bacteria in a single measurement. Multiplexing decreases the number of tests required for a complete analysis of a sample. This, in turn, decreases cost and workload.

We propose here an innovative low-cost label- and binding-free method with multiplexing capability that relies on microscopy imaging, image processing and machine learning in order to detect the proportions of a selection of bacteria in a sample. In this paper, the focus is put on four of the most important pathogens hand-related bacteria, namely *Escherichia coli, Staphylococcus aureus*, *Pseudomonas aeruginosa* and *Corynebacterium jeikeium*. The cultivation of bacteria, the optical set-up and the algorithms used for image processing, classification and machine learning are described in the Material and Methods section. Results are then presented for specific bacteria solutions first in water, then in isopropanol. Finally, the results and outlooks of the project are discussed.

## 2. Materials and Methods

### 2.1. Bacterial Strains and Cultivation

*Escherichia coli*, K12 strain, was cultivated in Luria Bertani broth; *Staphylococcus aureus* (clinical strain A170) was cultivated in Tryptic Soy Broth supplemented with 3% (w/v) yeast extract; *Pseudomonas aeruginosa* (clinical strain) was cultivated in Tryptic Soy Broth and *Corynebacterium jeikeium* (purchased from DSMZ, DSM-7171) was cultivated in Tryptic Soy Broth supplemented with 0.2% tween 80.

For simplification, the bacteria strains are referred in this paper as follows: EC, *Escherichia coli*; SA, *Staphylococcus aureus*; PA, *Pseudomonas aeruginosa*; CJ, *Corynebacterium jeikeium*.

For image acquisition, all bacteria were thawed and grown overnight at 37 °C (stationary phase). Concentration was then estimated by OD measurement.

### 2.2. Optical Set-up

Bacteria were diluted in either Phosphate Buffered Saline (PBS) or 70% isopropanol (v/v) to obtain a 10^7^–10^8^ bacteria/mL concentration. The solution flows through a microfluidic chip (µ-Slide I Luer from Ibidi, catalogue number: 80171) toward an imaging chamber. Pictures are taken with an Evos Fl microscope from AMG and a 100 × oil objective.

### 2.3. Image Analysis

Acquired images were further processed using a self-made Fiji-based macro [19]. First, Kuwahara filters [20] were applied to remove noise and the illumination background. Second, a simple segmentation is performed to identify objects in images and calculate, for each, seven morphological parameters (MPs). This step is called “segmentation” in the following. These MPs are described in Table 1. Each object might be an isolated bacterium or a cluster composed of several aggregated bacteria. Third, a watershed segmentation [21] was performed on each object to break the cluster and detect individual items inside each cluster. This step is called “declustering” in the following. For each cluster, seven average morphological parameters (AMPs) were computed by averaging the MPs of each item of the cluster. MPs, AMPs, as well as the number of items identified in each object are recorded in a Table, one per type of bacteria (see Figure 1). In the following, the term “object” refers to what the simple segmentation algorithm detects, the term “cluster” is used to design such an aggregate, the term “item” corresponds to individual bacterium identified inside a cluster by the declustering algorithm and the term “object parameters” (OPs) is used to designate the 15 parameters (MPs, AMPs, number of items) associated to an object.

### 2.4. Classifiers

Two of the most common supervised machine learning algorithms have been compared: the Random Forest (RF) and the Support Vector Machine (SVM). Additionally, three classification strategies have been implemented: the 1-over-4 which is the most straightforward, the binary strategy which relies on binary classifiers but may lead to indecision and the tournament classifiers that also rely on binary classifiers but is expected to be more efficient in overcoming indecision. Machine learning algorithms and classification strategies are described in the following subsections. The training and the classification are performed on tables of OPs provided by ImageJ. Several datasets have been built from this table: *N* learning datasets (LDSs) and *N* verification datasets (VDSs). Both are composed of an equal number of objects randomly picked in tables for each type of bacteria (see Figure 1). There are about three times more objects in the LDS than in the VDS. Unless specified, *N* = 50 for the analysis presented in this paper.

#### 2.4.1. Random Forest

RF is based on the principle of a decision tree (DT). In such a binary tree, each branch corresponds to a test on a given OP while each leaf is associated with a label [22]. The RF algorithm calculates multiple DTs on randomly drawn subsets for the LDS. An unknown object passes through each DT. Then the class predicted by the algorithm corresponds to the class predicted by the majority of DTs. The sklearn 0.20.4 Python module has been used to calculate this classifier. Three main parameters of the algorithm were adjusted to optimise the learning and classification process: the number of DTs in the forest, the maximal depth of the tree and the minimal number of objects of the LDS in each leaf.

#### 2.4.2. Support Vector Machine

SVM consists of calculating equations of hyperplanes that divided the space of OPs into subspaces regrouping the same bacteria [22]. An unknown object is given as input to this classifier simply receives a label according to its position in the OP’s space. Again, sklearn 0.20.4 Python module has been used to implement this classifier. Once more, three parameters of the algorithm were adjusted to optimise the classifier: the type of hyperplane equations we used (linear, polynomial, sigmoid, etc.), an associated coefficient and a regularisation parameter.

#### 2.4.3. 1-over-4 Classification Strategy

The 1-over-4 classification is a straightforward strategy that consists of a single classifier with four possible classes, one per bacterium to identify.

#### 2.4.4. Binary Classification Strategy

In the binary classification strategy, a binary classifier is calculated per bacterium. This classifier aims at determining if the object is the targeted bacterium or not. Thus, each classifier returns a Boolean value. Outputs of each classifier are then compared to predict a class. In the most favourable case, only one classifier returns “true” (single-hit) and the determination of the class of the unknown object is unambiguous. However, multiple classifiers sometimes return “true” on the same object (multiple-hit). In this case, the class is attributed to the object associated with a probability. This strategy is a bit more complex to implement. It requires as many classifiers as the number of bacteria. However, ambiguous cases can be easily detected and processed.

#### 2.4.5. Tournament Classification Strategy

For the tournament classification strategy, a set of binary classifiers is computed, one for each couple of labels. An unknown object passes through all the classifiers. For each classifier, the selected label earns one point. At the end of the process, the predicted label is the one with the highest number of points. Sometimes, several labels can end up tied at the top. In this case, the label is assigned with a certain probability, as in the case of the binary classifier. The strength of this approach is to rely on several classifications, which tend to smooth the impact of classification errors. However, this strategy is hardly scalable to a large number of bacteria: the number of required classifiers increases with the square of the number of bacteria.

### 2.5. Classifier Validation and Metric

Once computed, each classifier was validated on the VDS by comparison of the predicted and the actual label. Results were represented in two ways: the confusion matrix and the bacteria count table. All data presented in the paper have been averaged over 50 runs with 50 different randomly drawn LDSs and VDSs for each configuration (see Figure 1). The way how to interpret confusion matrices is described in Figure 2. Several results have to be taken into account to quantify the accuracy of a classifier:The overall score, on the bottom-right box, that gives the ratio of objects of the VDS for which the prediction matches the actual bacterium label.The individual score, on the bottom row, for each bacterium, which provides the ratio of appropriate labelling for a given bacterium.The bacteria count which should be as homogeneous as possible, considering that the number of each bacterium in the VDS is the same. If inhomogeneity occurs, the classifier is considered to be biased.

Confusion matrices are always given for the VDS but the classifier is also applied to the LDS. This procedure is used to fine-tune the parameters of the RF or the SVM algorithm to find the highest recognition accuracy while avoiding overfitting issues. Overfitting is like rote learning on the LDS, which can be quite inefficient on unknown objects. Balancing the scores obtained on LDS and VDS is a good way to avoid overfitting.

### 2.6. Datasets

Several sets of images have been realised for this study. First, we acquired two series of images in a PBS solution containing only one of the four studied bacteria. Acquisitions have been made independently. The first set of images, composed of about 30 images per bacterium, has been used to compute the classifier. The LDS and VDS have been built from these images data. The second set of images has been used to double-check the results.

The same methodology has been applied to the study in isopropanol. The first dataset is composed of about 50 images per bacterium while the second is composed of about 20 images per bacterium. In addition, for isopropanol, images on mixtures of several bacteria at different ratios have also been acquired.

## 3. Results

Image analysis results are presented in this section. The focus is first put on images acquired in PBS. The performances of image processing algorithms are shown. Then, results obtained with different learning and classification algorithms are compared. The same approach is applied to images acquired in isopropanol. Finally, the performances of the complete recognition of bacterial mixtures acquired in isopropanol are exhibited.

### 3.1. Image Processing of Images Acquired in PBS

The first results are obtained with bacteria in a PBS solution. Images of solutions that contain only one kind of bacterium have been taken, and the ImageJ script to detect objects and compute morphological parameters (MPs) has been applied to each of them. Examples of raw and processed images are depicted in Figure 3A. It is observed that each type of bacterium has a slightly different morphology, but the distinction between them is not straightforward by eye (e.g., *E. coli* and *P. aeruginosa*). Moreover, several bacteria form clusters. Most of the time, they do it voluntarily by quorum sensing and to form biofilms [23,24,25]. Cluster formation also depends on the hydrophobicity of the cell wall [26]. The pattern of the cluster is also characteristic of a given type of bacterium (for instance, *S. aureus* often forms a diplococcus, streptococci assemble to long chains, etc.).

The results of the segmentation and object labelling are presented in Figure 3C for *S. aureus* and Figure 3G for *C. jeikeium*. The outlines of each object are well detected whenever the contrast compared to the background is sufficient. Second, Figure 3D,H depict the results of the declustering algorithms for *S. aureus* and *C. jeikeium,* respectively. The declustering process is not ideal. For instance, in Figure 3B, two diplococci can be identified by the eye at the centre of the image. These two diplococci are respectively identified as an isolated object and as a cluster formed by three items in Figure 3D. However, even if the declustering process is not perfect, it still provides a quantitative metric of the tendency of a bacterium to form clusters and of the morphology of clusters. This information is complementary to MPs of isolated bacteria and might help for the classification.

MPs are computed on all objects of the segmented images and recorded in a table. Figure 4 illustrates the histograms and the fitted distribution of the seven MPs, as well as the number of items detected in clusters, for each type of bacteria. Data are normalised by the number of objects of each type in the complete data set. The histograms overlap with each other, which should make the classification trickier. However, some parameters seem to be rather discriminative. For instance, it is the case for the circularity.

### 3.2. Classification of Images Acquired in PBS

The purpose of the classifier is to assign a label to each object of an image according to the values of its MPs and the number of items per object. AMPs are not used for this part of the study. The complete dataset used to train and verify the classifier in PBS is composed of 489 objects for *C. jeikeium*, 560 for *E. coli*, 1000 for *P. aeruginosa* and 444 for *S. aureus*. The number of objects per type of bacteria in each LDS and VDS is respectively 311 and 133. The supervised machine-learning algorithms and classification strategies employed in this process are described in Material and Methods. Figure 5A,B depict respectively the confusion matrices for the 1-over-4 classifier with RF and SVM algorithms. The results are convincing. Despite the quite large overlap of histograms (see Figure 4), the overall classification score reaches 75%. The RF overall score is slightly better than SVM one, but the difference is not significant. However, the individual scores for each type of bacteria are more uniform with RF than with SVM: the range of individual scores fluctuates between 74% and 79% for RF and between 58% and 91% for SVM. For SVM, the individual scores highlight a lower prediction rate for *P. aeruginosa* compensated by an overestimation of the number of *C. jeikeium*. Confusion matrices for binary classifiers obtained with the RF algorithm are also presented in Figure 5C–F. The recognition rate for one type of bacterium against others is about 90%.

The classifier uses seven parameters but the relevance of all of them is questionable. To assess this point, we calculated classifiers based on subsets of MPs. The difference we measured between the best subsets of MPs and the full set of MPs is not so significant. Thus, in the following, the full set of MPs is considered. We also assessed the impact of the size *L* the LDS. It turns out that the overall score increases with *L* but saturates for *L* > 150 per type of bacteria. The size of our LDS, i.e., 311 per type of bacteria, is thus appropriate.

Finally, the three classification strategies are also compared. Table 2 is a simplified representation of an equivalent for the confusion matrices obtained from the 1-over-4 classifier, the binary classifier and the tournament classifier. It should be noted that results in Table 2 are slightly higher than in Figure 5 because the score in Table 2 is computed on the complete dataset. The tournament classifier beats the two other ones on the individual score for each bacterium. The binary approach is also worth of interest as it is the only strategy able to exclude objects (if none of the binary classifiers matches). In the case of the 1-over-4 or the tournament classifiers, all objects receive a label, even if their MPs are far from the nominal ones. This explains why the false-positive rates are higher for the 1-over-4 strategy (up to 10.9%, 7.1% on average) or the tournament strategy (up to 8.0%, 4.7% on average) than for the binary strategy (up to 7.5%, 3.5% on average).

### 3.3. Image Processing of Images Acquired in Isopropanol

The ultimate goal of this detection system is to analyse samples picked up in the discard after a hand-hygiene event. Thus, bacteria are no more in a PBS solution but rather in hydroalcoholic solution or any disinfectant solution. Commonly used are alcohol-based solutions, most of them being composed of isopropanol. Thus, we extended our study to a more realistic case, i.e., the analysis of bacteria in a solution of 70% *w*/*w* isopropanol.

Figure 6 shows pictures of the four bacteria acquired in isopropanol solution. Isopropanol seems to increase the propensity of bacteria to form clusters (see Figure 3 for comparison). This might be because alcohol dissolves the bacterial membrane, which is composed of fatty acids (lipids) and proteins. Thus, it reacts with the acidic part of fatty acids to form esters. As esters are hydrophobic whereas the acidic part is hydrophilic, the formed esters of the residual membranes have then the tendency to aggregate [27]. This phenomenon is likely to increase the difficulty of the recognition process.

We apply the same workflow as for the analysis of images acquired in PBS. Figure 7 shows the histograms and fitted distributions of the 15 OPs. Data are normalised by the number of objects of each type in the complete data set. The shape of the distribution of the MPs’ changes in comparison to the ones obtained for images acquired in PBS (see Figure 4). Fitted distributions are much wider and flatter in isopropanol than in PBS and the range of values reached by each MP is larger. This is probably a consequence of the clustering of bacteria by the alcohol.

### 3.4. Classification of Images Acquired in Isopropanol

First, we computed classifiers with the seven MPs of objects and the number of items identified inside objects by the declustering algorithm. Performances, which are not shown in this paper, were not convincing. The overall score was below 45 % and *S. aureus* had an individual score below 20 %. Then, we added the seven AMPs, increasing the number of parameters for this classification to 15. The histograms of these seven new parameters are also presented in Figure 7 over grey background. The distributions of these last seven parameters are close to the ones we had for images acquired in PBS. This is expected to improve the recognition process.

The complete dataset used to train and verify the isopropanol classifiers is composed 1537 objects for *C. jeikeium*, 4608 for *E. coli,* 2109 for *P. aeruginosa* and 1111 for *S. aureus*. The number of objects per type of bacteria in each LDS and VDS are respectively 750 and 250. The same strategies and supervised machine-learning algorithms have been used. Figure 8 depicts the resulting confusion matrices for the 1-over-4 strategy for RF and SVM, and the binary strategy for RF. Using OPs instead of MPs improves the quality of the classification process. However, the overall score for isopropanol remains lower than the one for PBS (56.2% against 76.7%). The individual score for each bacterium fluctuates between 46.4% and 72.8% for RF and 29.6% and 77.2% for SVM. In particular, there is a large error for the classification of *E. coli* with SVM method: half of them are classified as *P. aeruginosa*. Thus, it can be pointed out that RF is more efficient than SVM according to this criterion. In addition, the individual scores are less uniform for images acquired in isopropanol than for images acquired in PBS, even for RF.

With binary classifiers, the score is between 68.6% and 79.4%, which is again lower than the score obtained for images acquired in PBS (90%). The maximal ratio of false-positive prediction is 18.8% for *C. jeikeium* while the maximum ratio of false negative is 17.8% for *P. aeruginosa*. Table 3 compares the performances of the three strategies in the same way as performed for images acquired in PBS. The results obtained with the 1-over-4 and the tournament strategies are quite similar, and both are better than the binary strategy. For this strategy, the number of unclassified objects is very low in comparison with the results for images acquired in PBS. The widening of the MPs’ distribution might explain this fact: the range of parameters that are uncovered decreases, which in turn reduces the range of MPs for which no classifier matches.

Finally, the rates of single-hits (objects for which only one of the four classifiers matches), double-hits (2 classifiers match), triple-hits and quadruple-hits for the binary strategy in PBS and isopropanol are depicted in Table 4. This result confirms that recognition in isopropanol is more confused. More than 60% of the objects match with more than one classifier in isopropanol while this value is only 1.5% in PBS.

### 3.5. Mixture Analysis

For this last experiment, new solutions of pure bacteria and mixtures of bacteria have been analysed with the complete workflow, using the 1-over-4 RF for the classification process. The image set is composed of 15 images for each solution. The ratio of objects classified as *E. coli*, *C. jeikeium*, *S. aureus* and *P. aeruginosa* is computed for each image. Figure 9 illustrates, for each solution, the average value and the standard deviation of this ratio over the image set. For the pure solution (Figure 9A–D), the recognition is not perfect but the estimated ratio of the actual type of bacterium is at least twice the ratio of the others. *S. aureus* is the best case (0.71 versus 0.12) whereas *P. aeruginosa* is the worst case (0.43 versus 0.20). Such values are in accordance with the ones that can be expected regarding the confusion matrices in isopropanol, where *P. aeruginosa* and *S. aureus* had respectively the lowest and the highest individual scores (Figure 8).

Four mixtures were also analysed: one with *C. jeikeium* and *S. aureus* only (CJ/SA), one with *C. jeikeium*, *S. aureus* and *E. coli* (CJ/SA/EC) and two with different ratios of *P. aeruginosa* and *S. aureus*. (PA/SA#1 and PA/SA#2). For the first mixture (i.e., CJ/SA, Figure 9E), the results are consistent as the two types of bacterium with the highest score are the two used for the mixture. However, the ratio between the lower true-positive (0.32 for *C. jeikeium*) and the higher false-positive (0.21 for *S. aureus*) is quite low. A similar observation can be made on the second mixture (i.e., CJ/SA/EC, Figure 9F). We also analysed two mixtures with two different ratios of *S. aureus* and *P. aeruginosa* with a binary *P. aeruginosa*/*S. aureus* classifier. Results are presented in Figure 9G and H. The histogram of mixture PA/SA#1 (Figure 9G) suggests that the solution is composed of about 2 SA for 1 PA. Moreover, there is three times more PA in the mixture PA/SA#2 than in the mixture PA/SA#1. Thus, the mixture PA/SA#2 is expected to be composed of 3 PA for 2 SA. This is consistent with the values measured (Figure 9H).

## 4. Discussion

The results presented in this paper support that combining optical imaging and artificial intelligence is a versatile tool for the detection of modifications in the composition of bacterial mixtures in PBS but also in isopropanol. This technique paves the way for integrated detection systems in the field of hand hygiene, and more precisely in the detection of changes in hand microbiome. Up to now, this study only covers four components of the hand microbiome, namely *E. coli*, *C. jeikeium*, *S. aureus* and *P. aeruginosa*, and will be extended to other bacteria.

Even if the detection and classification of bacteria are still perfectible and do not perform as well as other techniques that were described in the introduction (PCR, SERS, SPR), the obtained results are convincing enough to consider in the future innovative applications in the field of infection prevention. The required set-up and the protocol are simple which should facilitate the integration of such a system into existing medical devices (e.g., hand sanitiser, see Figure 10) and make it a good candidate for a large-scale early-alert system. The major benefits of such a dispenser-integrated monitoring system are to run simultaneously to a hand hygiene event and does not take any extra time nor action to the user. As mentioned before, the method detects an increase of specific bacteria and gives the user a feedback for further medical assistance. Thus, our method is complementary with common detecting methods for detecting an infection that are in general all more cost- and time-consuming.

Future investigations will focus on three main aspects: the integration inside medical devices, the improvement of the identification algorithm and the transfer from lab condition experiments to real life. Regarding the first point, state-of-the-art technologies in microfluidics, microscopic imaging and electronics seem to be mature enough for the miniaturisation of the set-up and its integration into a hand sanitiser. A sketch of the final product we target is given in Figure 10.

Figure 10A represents a front view of an existing commercially available dispenser commonly used in the hospital environment and that is large enough to host our future integrated device. Figure 10B sketches a possible integration setup. The left compartment of the dispenser will be for the disinfectant tank, pumps and tubing. The pump activation sensory and associated electronics will be placed in the back of the central aperture for the hands. A Microfluidic device aiming to gathering and preparing samples for analysis will replace the drip tray located below the hands aperture. Finally, the right compartment of the dispenser provides enough space to integrate a miniaturised microscope, a microfluidic chip, batteries and embedded systems for data processing and communication.

In addition, we also identified some avenues to pursue in order to improve the detection efficiency. Our approach to break clusters and identify bacteria inside clusters has shown its limitations. Clustering malfunctions introduce errors in the data used to train the classifier as well as in the data used for the image analysis. This bottleneck definitely needs to be deeply investigated. The current image processing algorithm involves standard denoising and segmentation methods. However, more sophisticated algorithms already exist and have proved their efficacy [28,29,30]. However, we have to keep in mind that our goal was a portative device. Thus, the computation power and the energy required to run the algorithms are also a strong limiting factor that has to be considered.

The classification algorithm might also be improved. The comparison between RF and SVM highlights the advantage of the first one. Derivations of RF, such as gradient boosted decision trees, deserve to be considered [31]. Nevertheless, the distribution of morphological parameters suggests that the margin for improvement is quite small and that confusion seems unavoidable. However, this is not a strong limitation if the aim is to perform a differential comparison between several solutions rather the spectroscopy of a single solution.

Our identification approach relies on a separation between image analysis and classification. An alternative consists in identifying bacteria directly on images, without using the MPs, which has already been demonstrated for different imaging techniques (standard optical imaging, 3D phase imaging, radiography) [32,33,34]. In these papers, deep-learning algorithms are implemented. Again, such algorithms require larger computation power and are more time-consuming, which can be a strong limitation for embedded applications.

In parallel with the integration of the system into a hand sanitiser and the improvement of processing algorithms, further investigations have to be carried out in order to move from a proof-of-concept study performed in the laboratory towards a real-case prototype. These upcoming works include the following: (i) assessment of experimental conditions (e.g., temperature, growth state); (ii) the impact of an increase in the number of bacterial species (which tend to increase the image processing); (iii) the impact of shape change for a given pathogen that might be induced due to various environmental conditions or mutations (e.g., filamentation); (iv) the adaptation of the classifier to possible variations in the shape of a given pathogen; (v) the distinction between commensal and pathogenic bacteria strains; (vi) the study of the image processing algorithm on more complex samples and on hand-washing samples (which will probably include exclusion criteria for objects that are not pathogens).

The present system can detect and identify four bacteria in a given solution, here for instance in PBS or isopropanol. Although it was initially intended for hand-hygiene purposes, it could easily be extended to infection diagnostics and pathogen identification in body fluids, for example in the blood (sepsis detection) or urine. This would significantly shorten diagnostic delays and appropriate medication adaptations, which is particularly relevant in the case of sepsis where time can be crucial. Nonetheless, the identification of bacteria in the blood would remain challenging, given its complex composition. Our technology could also be extended to quality control of water or several products destined for consumption (e.g., milk, drinks).

## 5. Conclusions

In conclusion, we presented a suitable method for label-free optical detection of pathogens in alcohol-based solutions, especially isopropanol. It relies on label-free optical microscopy and machine learning algorithms. Compared to existing ones, our approach has four main advantages. First, it does not require prior labelling or binding of specific molecules. Second, it is a low-cost approach. Third, it can easily be integrated into several medical devices such as disinfectant dispensers. Fourth, it can be integrated and work in parallel with standard hand hygiene procedures. Thus, it does not require any extra time or effort by the user. Our pathogenic detection method has been validated on mixtures of four different pathogens, both in PBS and isopropanol. We showed that we can detect and identify bacteria with an accuracy of more than 50%. Within this, the method allows identifying single types of bacteria in an aqueous or alcohol-based solution. Several algorithms and approaches have been tested and it can be pointed out that the more suitable one is the Random Forest classification algorithm coupled with a 1-over-4 classification strategy. This method seems therefore to be suitable for the development of a system that will be able to detect unusual concentrations of pathogens on the hands during the disinfection process, which would be a major milestone in infection prevention.

## Figures and Tables

**Figure 1 biosensors-11-00002-f001:**
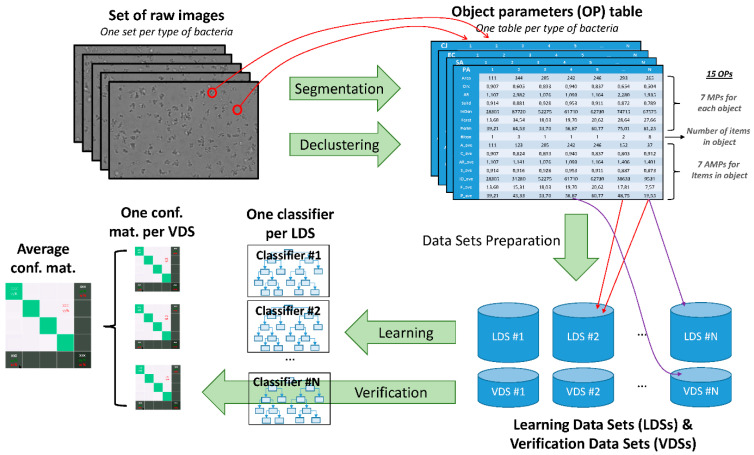
Description of the complete workflow. First, a set of images is acquired for each type of bacteria. Second, image-processing algorithms, namely the segmentation and the declustering are applied on images in order to obtain a table of parameters for each type of bacteria. The segmentation provides the seven morphological parameters (MPs) while the declustering provides the number of items in each object and the seven average morphological parameters (AMPs). Third, *N* learning datasets (LDSs) are prepared by picking randomly a same number of columns in each table. *N* verification datasets (VDSs) are prepared by the same way. There are about three times more objects in the LDS than in the VDS. A same column can be used for two different LDS, or for an LDS and another VDS but not for both an LDS and its associated VDS. Fourth, machine-learning algorithms are applied on each LDS to obtain *N* classifiers. Fifth, classifiers are applied to the corresponding VDS to compute the *N* confusion matrices. Results shown in the paper correspond to the average result over the *N* dataset.

**Figure 2 biosensors-11-00002-f002:**
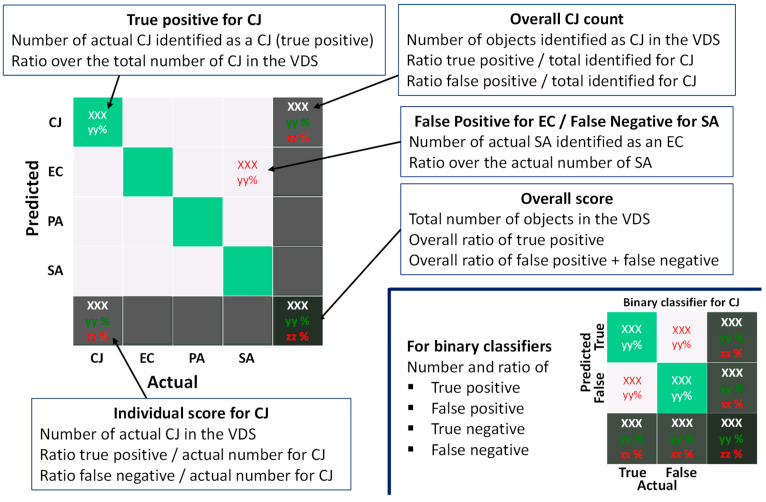
Description of the way how confusion matrices have to be interpreted in the case of binary classifiers (bottom-right) or the 1-over-4 classifier.

**Figure 3 biosensors-11-00002-f003:**
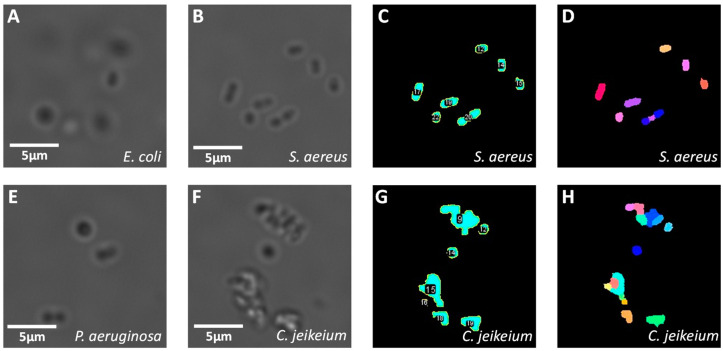
Results of image processing for acquisitions made in PBS. (**A**,**B**,**E**,**F**) are respectively zoomed images of *E. coli*, *S. aureus*, *P. aeruginosa* and *C. jeikeium*. (**C**,**G**) are the binary masks used to identify and label objects in (**B**,**F**), respectively. (**D**,**H**) are the labelled images computed by the declustering algorithm. Colours are arbitrarily chosen and indicate what the algorithm identifies as objects (i.e., individual bacterium or clusters) in (**C**,**G**), and as items (individual bacterium), whether it belongs to a cluster or not, in (**D**,**H**). Scale bars: 5 µm.

**Figure 4 biosensors-11-00002-f004:**
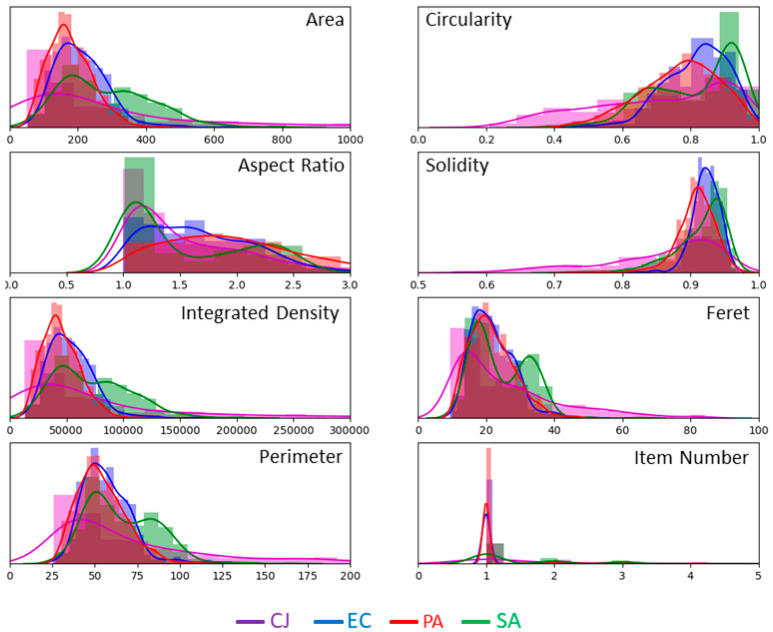
Histograms and fitted distribution of MPs computed on the images acquired in PBS. Plots are normalised, for each type of bacteria, by the number of objects in the dataset, i.e., 511 for *C. jeikeium*, 519 for *E. coli*, 917 for *P. aeruginosa* and 419 for *S. aureus.*

**Figure 5 biosensors-11-00002-f005:**
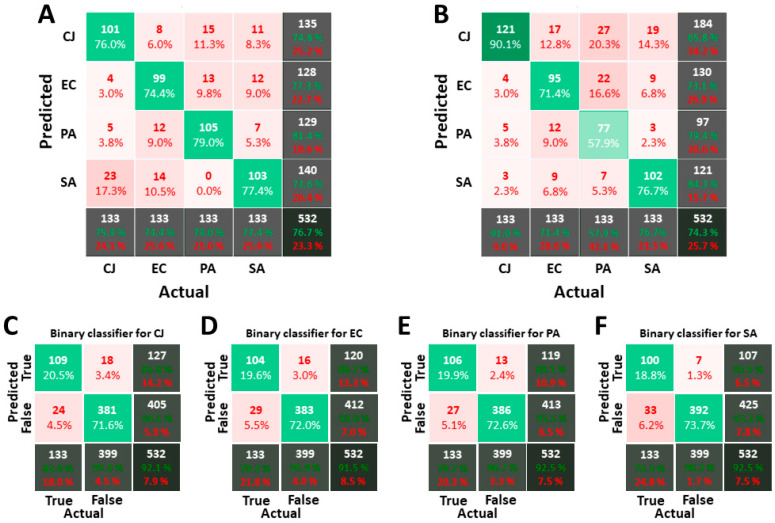
Confusion matrices for classifiers calculated on images acquired in PBS. (**A**,**B**) are the confusion matrices obtained with the 1-over-4 strategy and Random Forest (RF) for (**A**) and Support Vector Machine (SVM) for (**B**). (**C**–**F**) are respectively the binary classifiers for *C. jeikeium*, *E. coli*, *P. aeruginosa* and *S. aureus*. The way to interpret confusion matrices is explained in Material and Methods. For the 1-over-4 strategy, the standard deviation over 50 runs is about 1.3% on the overall score for RF and 1.5% for SVM, and about 3.5% on individual scores for RF and 3.2% for SVM. For the binary strategy, the standard deviation is about 1% for the overall score for RF and 1.2% for SVM, and 3% for the individual score of true positive for RF and 1.5% for SVM.

**Figure 6 biosensors-11-00002-f006:**
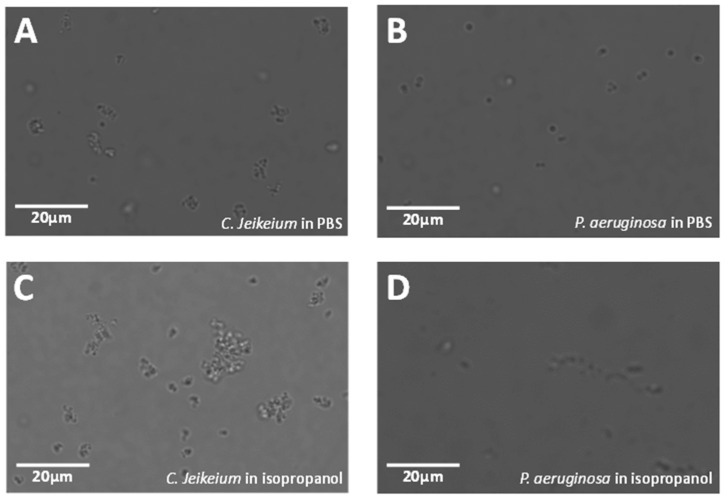
Comparison of images of *C. jeikeium* and *P. aeruginosa* in PBS and isopropanol. (**A**,**C**) present *C. jeikeium* in PBS and in isopropanol, respectively. (**B**,**D**) depict *P. aeruginosa* in PBS and in isopropanol, respectively. Scale bars: 20 µm.

**Figure 7 biosensors-11-00002-f007:**
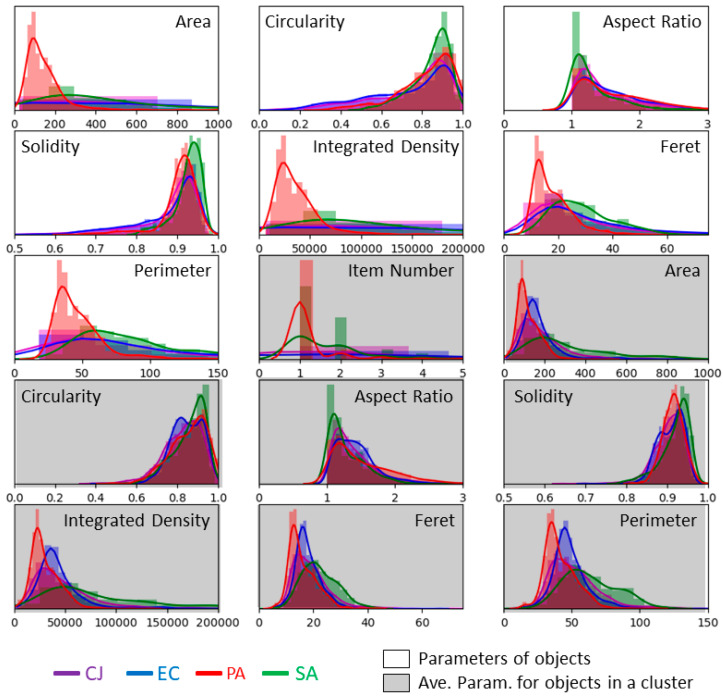
Histograms and fitted distributions of OPs computed on the images acquired in isopropanol. Histograms and fitted distributions of the seven MPs of objects (isolated bacterium or clusters), the number of items in each object and the AMPs of clusters are represented. Parameters with a white background correspond to MPs while parameters with a grey background correspond to the AMPs. Plots are normalised for each type of bacteria by the number of objects in the dataset, i.e., 2016 for *C. jeikeium*, 2439 for *E. coli*, 957 for *P. aeruginosa* and 324 for *S. aureus.*

**Figure 8 biosensors-11-00002-f008:**
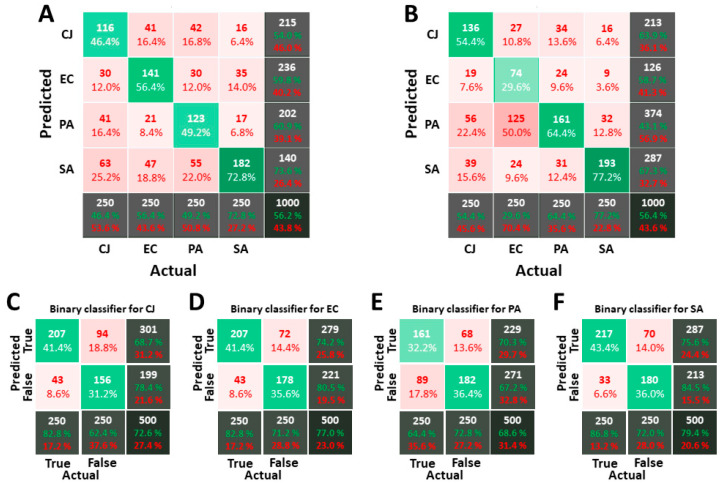
Confusion matrices for classifiers calculated on images acquired in PBS. (**A**,**B**) are the confusion matrices obtained with the 1-over-4 strategy and RF for (**A**) and SVM for (**B**). (**C**–**F**) are respectively the binary classifiers for *C. jeikeium*, *E. coli*, *P. aeruginosa* and *S. aureus*. The way to interpret confusion matrices is explained in Material and Methods. For the 1-over-4 strategy, the standard deviation over 50 runs is about 2.3% on the overall score for RF and 3.7% for SVM and about 3.8% on individual scores for RF and 4.1% for SVM. For the binary strategy, the standard deviation is about 1.2% for the overall score for RF and 2.1% for SVM and 3.7% for the score of true positive for RF and 3.5% for SVM.

**Figure 9 biosensors-11-00002-f009:**
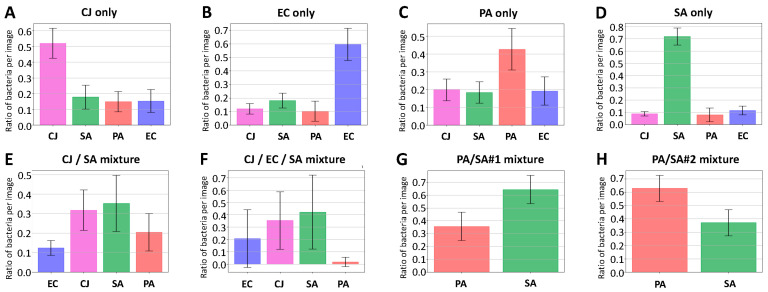
Analysis of mixture of bacteria. (**A**–**D**) correspond to pure solutions of *C. jeikeium*, *E. coli*, *P. aeruginosa* and *S. aureus*, (**E**) to a mixture of *C. jeikeium* and *S. aureus,* (**F**) to the same mixture with *E. coli* in addition and (**G**,**H**) to two mixtures of *S. aureus* and *P. aeruginosa* with different ratios (three times more *P. aeruginosa* in H). Images of all pictures have been analysed with RF and the 1-over-4 strategy, except for (**G**,**H**) that have been analysed with a RF *P. aeruginosa*/*S. aureus* classifier. Error bars correspond to the standard deviation of the ratio of bacteria measured over 15 images for each mixture. The average number of bacteria on each image is 30 ± 10 for (**A**), 150 ± 50 for (**B**), 150 ± 20 for (**C**), 120 ± 20 for (**D**), 25 ± 5 for (**E**,**F**), 60 + 20 for (**G**,**H**). Imaged bacteria are all in the stationary phase.

**Figure 10 biosensors-11-00002-f010:**
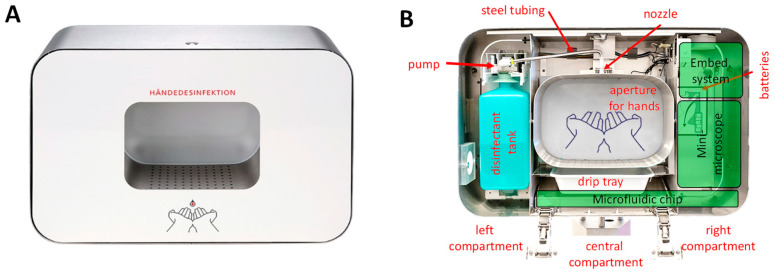
Commercially available dispenser as a platform for a future system integration of our method. (**A**) The front view of the dispenser. The dispenser has a compartment in which the hand-hygiene event takes place. The overall dimension of the dispenser is 337 mm × 506 mm. (**B**) An internal view of the dispenser. The disinfectant tank with the pump is located on the left compartment. The pump is connected with the middle compartment by a steel tubing. At the end of this steel tubing, the nozzle extract the disinfectant in the palm of the hands (central compartment). Below there is a drip tray that could collect the so-called overspray that might contain pathogens. Batteries are located on the right compartment. As there is still enough space, in the future, the drip tray could be replaced by a microfluidic chip to collect and prepare the samples and a miniaturised microscope, microfluidic pumps and an embedded system for data processing could be integrated in order to have everything dispenser-integrated (green boxes in the figure). It is noteworthy that this commercially available dispenser can be equipped with a Bluetooth module or with a power adapter.

**Table 1 biosensors-11-00002-t001:** Description of the morphological parameters used for the identification of bacteria.

Morphological Parameter	Symbol	Definition
Area	A	The number of pixels inside the detected object.
Perimeter	P	The number of pixels that compose the boundary of the object.
Circularity	C	Defined by the equation $C=4\cdot \pi \cdot A/P^{2}$.
Aspect Ratio	AR	The ratio of the major axis and the minor axis of the ellipse that fits the object outline. AR = 1 means that the object is perfectly round. AR tends to 0 for skinny objects.
Solidity	S	Defined as the ratio between the area of the object and the area of the smallest convex shape that encompasses the object. S = 1 defines a perfectly convex object. S tends to 0 for rough objects or objects with erratic outlines.
Integrated Density	D	The sum of the intensity of the pixels inside the detected object.
Feret diameter	F	The maximal distance between two points of the outline of the detected object.

**Table 2 biosensors-11-00002-t002:** Comparison of the individual score of each type of bacterium for images acquired in PBS for the three classification strategies: 1-over-4, binary, tournament. The cell background colour varies from red to green through orange and yellow depending on the percentage of bacteria correctly identified. The colourbar is not linear. Red corresponds to percentages below 5%, orange around 10%, yellow around 20% and green above 30%.

Actual	Bacteria in	Predicted	1-over-4		Binary		Tournament	
Label	the Dataset	Label	Number	%	Number	%	Number	%
CJ	489	CJ	378	77.3	365.5	74.7	413.3	84.6
		EC	29	5.9	14	2.9	21.3	4.4
		PA	34	7.0	17	3.5	27.3	5.6
		SA	48	9.8	15.5	3.2	27	5.4
		Others	0	0	77	15.7	0	0
EC	560	CJ	34	6.0	9	1.6	21.8	3.9
		EC	427	76.3	417	74.5	462.3	82.6
		PA	50	8.9	27.5	4.9	40	7.1
		SA	49	8.8	27.5	4.9	35.8	6.4
		Others	0	0	79	14.1	0	0
PA	1000	CJ	75	7.5	29	2.9	79.6	8.0
		EC	109	10.9	75	7.5	67.6	6.8
		PA	804	80.4	785	78.5	842	84.3
		SA	12	1.2	5	0.5	10	1.0
		Others	0	0	106	10.6	0	0
SA	444	CJ	32	7.2	16.5	3.7	12.6	2.8
		EC	33	7.4	17.5	4.0	16.6	3.8
		PA	18	4.0	9	2.0	5	1.1
		SA	361	81.3	369	83.1	409	92.2
		Others	0	0	32	7.2	0	0

**Table 3 biosensors-11-00002-t003:** Comparison of the individual scores of each type of bacterium in isopropanol for the Table 1. over-4, binary, tournament. The cell background colour varies from red to green through orange and yellow depending on the percentage of bacteria correctly identified. The colourbar is not linear. Red corresponds to percentages below 5%, orange around 10%, yellow around 20% and green above 30%.

Actual	Bacteria in	Predicted	1-over-4		Binary		Tournament	
Label	the dataset	Label	Number	%	Number	%	Number	%
CJ	1537	CJ	798	51.9	798	51.9	796	51.8
		EC	217	14.1	222	14.5	231	15.0
		PA	197	12.8	272	17.6	231	15.0
		SA	325	21.2	237	15.5	279	18.2
		Others	0	0	8	0.5	0	0
EC	4608	CJ	609	13.2	830	18.0	599	13.0
		EC	2890	62.7	2352	51.1	2751	59.7
		PA	257	5.6	613	13.3	474	10.3
		SA	852	18.5	782	17.0	782	17.0
		Others	0	0	29	0.6	0	0
PA	2109	CJ	439	20.8	540	25.6	455	21.6
		EC	356	17.0	316	15.0	345	16.4
		PA	867	41.1	916	43.4	890	42.2
		SA	444	21.1	326	15.5	417	19.8
		Others	0	0	11	0.5	0	0
SA	1111	CJ	78	7.0	164	14.8	105	9.4
		EC	133	12.0	135	12.1	135	12.2
		PA	81	7.3	103	9.2	73	6.7
		SA	819	73.7	704	63.4	796	71.7
		Others	0	0	5	0.5	0	0

**Table 4 biosensors-11-00002-t004:** Number of hits per objects for images acquired in PBS and in isopropanol for the binary strategy.

	No Hit	Single Hit	Double Hit	Triple Hit	Quadruple Hit
**PBS**	12.1%	86.4%	1.3%	0.0%	0.0%
**Isopropanol**	0.7%	39.1%	50.4%	9.7%	0.0%

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
