# Peer review of "A Label-Free Optical Detection of Pathogens in Isopropanol as a First Step towards Real-Time Infection Prevention"

_biosensors, 2020, doi:10.3390/bios11010002_

Round 1
Reviewer 1 Report
Overall Summary:
Julie Claudinon et al. describe an optical-based approach combined with machine learning to differentiate 4 bacterial strains (Escherichia coli, Staphylococcus aureus, Pseudomonas aeruginosa and Corynebacterium jeikeium) in PBS and isopropanol. The authors are motivated to develop a simple and label free approach to detect pathogens in order to prevent nosocomial infections. They aim to detect pathogens in isopropanol so that their detection system can potentially be coupled with hand sanitizing stations as a monitoring system. The authors mention that this can easily be achieved, however, the workflow for such a device is not directly apparent to me. Typically, when one sanitizes their hands, they do not produce eluent that can be collected and analyzed optically. Though the concept and motivation are rather creative and innovative, the data is not very convincing in its current form. For example, the number of bacteria analyzed when testing the model is not shown, the # of replicate experiments, nor error bars/confidence intervals.
Specific comments:
- Figure 1, I suggest wording ‘well-detected’ to ‘true positives’.
- Figure 2, Please mention what the colors are. For example, colors are arbitrary and indicate what the algorithm identifies as individual bacteria. What state were the bacteria for this experiment (i.e., exponential growth, stationary…). How does growth state change the results of training and indentification?
- Figure 3, Label y axis with counts so that the reader knows how many bacteria were analyzed to train the ML algorithm. Also, what exactly is object number?
- Line 183, The authors mention 40 images being analyzed, yet in line 232, the authors mention 444 objects analyzed?
- For all the precents and comparisons mentioned throughout the text, please include STDEV and/or confidence intervals.
- Line 243, what makes 90% satisfactory? (i.e. What is the threshold and how was this determined?)
- Figure 4A. It makes more sense to discuss about binary classifiers first before 1 over 4 analysis. My interpretation of binary analysis is that the algorithm is just trying to determine if the unknown bacterium is of a single strain, where for the 1 over 4 analysis, the algorithm is trying to distinguish 4 strains. It makes sense to talk about the simpler approach first. However, I may be misinterpreting these two analyses. Also, can the authors reformat the figure to show a heat map to better represent the data. It is very difficult to compare data.
- Can the authors please discuss a little why the three different ML algorithms were chosen. Are these the best 3, exc.?
- What leads to an ~75% success rate for RF 1-4 strategy? If the training data set were larger, can this be improved. The authors should compare confusion tables for ML trained on varying number (n) of datasets, and plot the % positives, % false positives, and % false negatives as a function of n.
- Line 251- Ferret ‘diameter’
- Line 250-255, How were the subsets created. Were all subsets measured, only combinations of three, or only the combinations you discuss. I suggest deleting line 250-255 and replacing with a single sentence saying that the data was subsampled but the results were not improved.
- Can you make heat maps of the confusion table and for Table 2. It would be much easier to compare the methods.
- Section 3.3 needs an introduction motivating why the authors are analyzing in isopropanol. The introduction is a bit abrupt.
- Figure 6, Use same vocabulary for Figure 6 (isolated and cluster, instead of objects and average).
- Figure 6. I suggest reorganizing the figure so that you can compare white versus gray. Very difficult to do so with current organization. Also, it is not very clear what is white and gray.
- Line 284 What is “Erreur! Source du renvoi”
- Line 302, I am confused how using the average parameter for a bacterium to the number of items inside a cluster increases parameters to 15.
- Line 312-313, Can the authors propose why SVM does not perform well on E. coli?
- Figure 8, What are the error bars? Also, how many bacteria were analyzed for each experiment. Were bacteria in log growth, exc. Please include this information in the figure description.
- Line 385, If data is not shown, please refrain from making any conclusions.
- Line 426, How would your method be used to detect viruses if they are below the detection limit. Please delete this claim or suggest how this wouldn’t limit your analysis.
- Line 434, How can it be integrated with medical devices?
Reviewer 2 Report
In this manuscript titled “Towards a label-free optical detection of pathogens in isopropanol in the field of infection prevention” Claudinon et al describe a novel method for detection of pathogenic bacteria using microscopic imaging and machine learning algorithms. They suggest that this approach could be adopted as a low-cost, label-free, and easy to integrate in any suitable medical device, such as hand hygiene dispensers. The rationale and the need for this approach is laid out well in the background and the authors have designed and conducted the study well and presented the data well. Having said that. the major criticisms that this reviewer has are: 1) Although it is a method that can be reasonably established to distinguish bacteria having distinct morphologies, such as the 4 bacteria used here, the problem is: training the algorithm with every possible pathogen that one will encounter in a nosocomial infection. 2) the second problem is that the same pathogen say E. coli can assume different shapes and sizes in different environmental conditions or mutation resulting in AMR- for example filamentation due to mutations that cause resistance to DNA damaging agents, 3. Also, the same pathogen (strain) exists in various shapes in human tissues and organs, e.g., Pseudomonas aeruginosa in lungs, 4) how this method can distinguish a commensal and pathogenic strain of the same strain for example VRE vs VSE. These are the pathogens of concern in nosocomial settings. 5) Finally, the feasibility of this approach in complex samples that you would normally see in hospital settings such as surfaces and hands etc. 6. This study primarily uses cultured pure samples and has not been tested in a complex mixtures.
Having said all these, this reviewer thinks that this is a good first step toward an approach that needs to be developed, refined and reported for acceptance and adoption. I am little disappointed that even the best results only reach about 70-80% true positive rate in this approach. So, how this will compare and compete against techniques such as portable PCRs which are have sensitivity and specificity reaching 95-99%? I strongly recommend that these critical limitations are pointed out and discussed in the discussion section and the proposition that this is the method that is better than currently available methods (mentioned in Introduction and other places in the manuscript) should be down played a bit to account for these limitations. But propose this as an alternate method in certain scenarios- at least to distinguish distinct morphotypes and species.
Some minor comments: I suggest taking a thorough look at the usage or words and context, syntax throughout the manuscript. Also, there are many places where references are messed due to formatting (shows error message).
- In the title: ‘field of infection control’ - sounds like in the field test- you really mean 'discipline' when you say field. I suggest rewording ‘as a tool for infection control’ or something to that effect
- Line 22: change ‘on four bacterium’ to ‘using four bacteria’
- Introduction - Good description of the state of the art and introduction of the concept
- Line 49: ‘needs specialised expertise and equipment’.This is not true anymore- there are faster, cheaper and potentially field deployable PCR devices.
- Line 51: sequencing required in normal PCR: Not sure what you mean by sequencing required- in end point PCR, if you use primers targeting specific pathogens and if you get the expected bands, then where is the need for sequencing? I don’t know if anyone uses PCR followed by sequencing of the amplicon as a detection method. I don’t even think that anyone uses end point PCR for detection/diagnosis. I suggest either removing this part of the sentence or reword.
- Line 91: detect alterations of a given microbiota- Again, I don't understand the concept of detecting alterations of a given microbiota. Known mixture of four bacteria does not represent microbiota. And the approach described here has not even been tested in appropriate matrices such as sputum or surfaces swab samples or hand wash liquid.
- Line 121 and many other places: The abbreviations such as LDS and VDS are not described anywhere. At least expand these at the first use.
- I definitely like the ML algorithm approach and it has been tried in the past and has been published – may not be in the same context. Again, my disappointment is that the best data on True positive is around 85%. Is that good enough for the nosocomial infection control scenarios? Please discuss what and where this is good enough for use compared to existing control measures in hospitals.
Round 2
Reviewer 1 Report
The authors have addressed most of my concerns and the manuscript is much improved. However, there is still need for some clarification.
- I thank the authors for including error in the figure caption. I still think it would be clearer to have error listed in the figure, but I leave this up to the authors and editor to decide.
- Regarding error, it is still not clear to me what the error is? Is the error from running the same dataset 50 times using the ML analysis? Or is the error from running 50 different experiments from different cultures? From the authors wording, it seems that error is from reanalyzing the same bacterial population 50 times. Could the authors provide details of what is the total # of bacterial versus subsampling. For example, Figure 4A shows ~140 CJ bacteria for this analysis. Does this mean 140 CJ bacteria were analyzed in total, and the ~1% error comes from 50 subsets of this culture (so an average # of 3 bacterium per 'experiment'). Or is the error from analyzing ~140 bacteria from a population of 'X'.
- The authors describe their experiments as the following: "All data presented in the paper have been averaged over 50 runs with 50 different randomly 188 drawn LDSs and VDSs for each configuration"
- Does this mean that for each experiment, the ML algorithm is retrained?
- What does 50 different randomly 188 drawn mean?
- Regarding increasing the # of parameters to 15 by using the average molecular parameters (MP) for objects in a cluster.
- This still is not fully clear. Does this mean the ML algorithm runs the analysis without declustering, and then runs a secondary analysis declustering using the same MP but adding a new Item # parameter along with the same seven MP? Maybe change nomenclature to training parameter when describing total parameters (i.e., 15).
- The authors more detailed discussion on how they envision the technology greatly improved the manuscript. I thank the authors for including this.
